# Enhancing Tumor Immunity with IL-12 and PD-1 Blockade: A Strategy for Inducing Robust Central Memory T Cell Responses in Resistant Cancer Model

**DOI:** 10.3390/antib13040094

**Published:** 2024-11-20

**Authors:** Fentian Chen, Kexin Wu, Shiqi Lin, Jinlong Cui, Xiaoqing Chen, Zhiren Zeng, Na Yuan, Mujin Fang, Xue Liu, Yuanzhi Chen, Wenxin Luo

**Affiliations:** 1State Key Laboratory of Vaccines for Infectious Diseases, Xiang An Biomedicine Laboratory, School of Public Health and School of Life Sciences, Xiamen University, Xiamen 361102, China; 32620170154850@stu.xmu.edu.cn (F.C.); liuxue1108@xmu.edu.cn (X.L.); 2State Key Laboratory of Molecular Vaccinology and Molecular Diagnostics, National Institute of Diagnostics and Vaccine Development in Infectious Diseases, Collaborative Innovation Center of Biologic Products, National Innovation Platform for Industry-Education Integration in Vaccine Research, Xiamen University, Xiamen 361102, China

**Keywords:** immune checkpoint, interleukin-12, PD-1, CTLA-4, central memory T cell

## Abstract

**Background:** Although immune checkpoint inhibitors (ICIs) have demonstrated efficacy in treating advanced cancers, their therapeutic success remains limited for many patients, with initial responders often experiencing resistance and relapse. Interleukin-12 (IL-12) is a powerful cytokine for antitumor immunotherapy, enhancing both lymphocyte recruitment into tumors and immune cell activation. **Methods:** In this study, we successfully produced mouse interleukin-12 (mIL12) through eukaryotic recombinant expression. In vivo, mIL12 exhibited significant control of tumor immunity in ICI-resistant and aggressive tumor models. Further mechanistic analysis indicated that treatment with mIL12 led to a substantial increase in tumor-infiltrating CD4^+^ T, CD8^+^ T, cDC1, and CD103^+^ cDC1 cells. **Results:** Our data underscore the potential of a combined therapeutic strategy involving IL-12 with PD-1 and CTLA-4 blockade to elicit a potent antitumor immune response. Notably, the co-administration of mIL12 and PD-1 blockade significantly enhanced the presence of central memory T cells (T_CM_) within tumors. **Conclusions:** This study is the first to provide evidence that the combination of mIL12 and PD-1 blockers promotes the generation of T_CM_, potentially contributing to a robust and durable antitumor effect.

## 1. Introduction

Tumor immunotherapy has emerged as a preeminent approach for cancer treatment. The first immune checkpoint inhibitor (ICI) was approved in 2011 for the treatment of unresectable melanoma following conventional therapies. Over the past decade, ICIs have revolutionized cancer treatment, becoming a primary therapy for various cancers. Current ICIs target proteins including programmed cell death protein 1 (PD-1), programmed cell death ligand 1 (PD-L1), cytotoxic T-lymphocyte-associated protein 4 (CTLA-4) and lymphocyte-activation gene 3 (Lag3) [1,2,3]. Despite widespread use, particularly of PD-(L)1 inhibitors, many patients do not achieve sustained antitumor responses, and issues such as drug resistance and relapse remain prevalent. Furthermore, a significant number of patients with positive PD-L1 expression do not mount effective immune responses, underscoring the urgent need for novel immunotherapeutic strategies or combination therapies involving ICIs [4,5].

The antitumor immune response is predominantly mediated by CD8^+^ T cells, with memory CD8^+^ T cells playing a pivotal role in sustaining long-term antitumor immunity. These memory cells are classified into four main subsets: central memory T cells (Tcm), effector memory T cells (Tem), stem cell memory T cells (Tscm), and resident memory T cells (Trm) [6,7,8,9]. Among them, central memory T cells (Tcm) are a type of circulating CD8^+^ memory T cell known for their high proliferative capacity and potent antitumor activity. Notably, Tcm cells have the ability to differentiate into both Tem and Trm subsets, further bolstering the antitumor response [10]. Clinical studies have demonstrated a positive correlation between the presence of Tcm cells and favorable outcomes in tumor immunotherapy, emphasizing the importance of inducing tumor-specific memory T cells for achieving durable antitumor immunity [11].

Additionally, previous research has shown that the antitumor efficacy of PD-1 inhibitors is dependent on IL-12 secretion by dendritic cells. Enhancing IL-12 secretion can improve the effectiveness of PD-1 inhibitors, suggesting that a combination therapy involving IL-12 cytokine and PD-1 inhibitors may represent an effective strategy for enhancing cancer immunotherapy [12]. Interleukin-12 (IL-12), a heterodimeric cytokine comprised of IL-12p40 and IL-12p35 subunits linked by a disulfide bond, plays a pivotal role in both innate and adaptive immunity [13]. Produced predominantly by antigen-presenting cells like dendritic cells and macrophages, IL-12 initiates signaling through the IL-12Rβ1 and IL-12Rβ2 receptor complexes, activating the JAK-STAT pathway. The production of IL-12 is influenced by various factors, including the regulatory mechanisms of IL-12 gene expression, the expression patterns of toll-like receptors (TLRs), interactions among different dendritic cell subsets, and the effects of cytokines such as interleukin-10 (IL-10) and type I interferon. IL-10 is known to negatively regulate IL-12 production by inhibiting the transcription of the genes encoding IL-12 [14,15]. IL-12 is a potent pro-inflammatory cytokine that not only induces the production of interferon-gamma (IFN-γ) but also promotes the differentiation of helper T cells into type 1 (T_H_1) [16]. This cytokine stimulates T and natural killer (NK) cells to produce a range of cytokines, including IFN-γ, granulocyte-macrophage colony-stimulating factor (GM-CSF), and tumor necrosis factor (TNF). It also facilitates IFN-γ production in conjunction with low levels of TNF and interleukin-1 (IL-1) [16]. Furthermore, IL-12 enhances T cell and NK cell activation synergistically with other cytokines and stimulates the T cell receptor-CD3 complex and immune complex formation with interleukin-2 (IL-2), leading to efficient IFN-γ production [17]. It enhances the cytotoxic activities of T and NK cells, promotes their production of IFN-γ, and stimulates the secretion of chemokines such as CXCL9 and CXCL10, which recruit effector cells into the tumor microenvironment [18,19,20]. In clinical trials, recombinant human IL-12 has been tested against various advanced malignancies, including renal cell carcinoma, melanoma, and non-Hodgkin’s lymphoma [21,22]. Moreover, recent preclinical studies have explored novel therapeutic modalities using IL-12, such as mRNA therapies, oncolytic viruses, CAR-T cells, and fusion proteins [23,24,25,26,27].

The potent antitumor effects of IL-12 are well-established. However, its clinical application has been limited due to dose-limiting immune-related adverse events (irAEs), such as cytokine release syndrome and liver injury [21,28,29,30,31,32]. These toxicities have hindered the approval of IL-12 for clinical use despite its considerable promise in cancer therapy. Consequently, devising methods to safely and effectively deploy IL-12 in clinical settings poses a substantial challenge. Potential strategies to overcome these hurdles could include optimizing dosing regimens, employing combination therapies, or developing modified IL-12 molecules that exhibit reduced toxicity, thereby maximizing the therapeutic potential of IL-12 while minimizing its adverse effects.

This study evaluates the antitumor efficacy of mouse IL-12 (mIL12) in several immunosuppressive mouse tumor models, demonstrating that mIL12′s antitumor activity is dose-dependent. Mechanistic analyses suggest that mIL12 remodels the tumor microenvironment into a pro-inflammatory state and upregulates the expression of PD-1 and CTLA-4. Additionally, when combined with PD-1 and CTLA-4 inhibitors, mIL12 enhances the effect, significantly increasing the population of tumor-infiltrating central memory T cells. These findings propose a promising new strategy for integrating IL-12 with immune checkpoint inhibitors in cancer therapy.

## 2. Method

### 2.1. Cell Line and Cell Culture

MC38 cells were acquired from the China Infrastructure of Cell Line Resources. CT26 and B16-F10 cells were sourced from the American Type Culture Collection (ATCC). All cell lines, including MC38, CT26, and B16-F10, were cultured in Dulbecco’s Modified Eagle Medium (Thermo Fisher Scientific, Walthan, MA, USA) supplemented with 10% fetal bovine serum (Thermo Fisher Scientific). EMT6 cells, also from ATCC, were maintained in Waymouth’s MB 752/1 medium (Thermo Fisher Scientific) containing 10% fetal bovine serum. Expi 293F^TM^ cells were purchased from Thermo Fisher Scientific and cultured in KOP 293 medium (Kairui Biotech, Zhuhai, China).

### 2.2. Cell Counting

The cells were resuspended to ensure even distribution, and 20 μL of the cell suspension was mixed with 20 μL of Trypan Blue (Merck Sigma-Aldrich, Darmstadt, Germany). Subsequently, a 20 μL aliquot of this mixture was placed into a counting chamber. Cell counting and viability assessment were performed using a Counterstar IC1000 cell counter (Conterstar, Shanghai, China), which provided both cell counts and viability metrics.

### 2.3. Expression and Purification of mIL12

The gene encoding mouse IL-12 (mIL12) was cloned into the eukaryotic expression vector PTT5 (Youbio, Changsha, China). This construct was transfected into Expi293F^TM^ cells using the KOP293 Transfection Kit (Kairui Biotech) following the manufacturer’s protocol. After 7 days, the cell supernatant was harvested, and the protein was purified. The mIL12 protein was purified via Ni-NTA chromatography (GE Healthcare, Chicago, IL, USA) according to the manufacturer’s instructions. Elution was carried out using 250 nmol/L imidazole, followed by dialysis of the eluted protein against PBS. The quantification of mIL12 was performed using a Multiskan GO microplate spectrophotometer (Thermo Fisher Scientific).

### 2.4. Sodium Dodecyl Sulfate-Polyacrylamide Gel Electrophoresis (SDS-PAGE)

mIL12 (3 μg) was mixed with either reducing or non-reducing loading buffer to achieve a final volume of 15 μL, supplemented with PBS. The samples were boiled for 10 min and then subjected to separation on precast polyacrylamide gels GenScript. Proteins were stained using eStain^®®^ L1 protein stain (GenScript, Piscataway, NJ, USA) for 10 min and visualized with a FUSION FX Spectra imaging system (VILBER BIO IMAGING, Paris, France).

### 2.5. Measurement of IFNγ by In Vitro Stimulation

Spleens were harvested from mice, homogenized, and treated with RBC lysis buffer (Solarbio, Peking, China) to obtain mouse splenocytes. A total of 5 × 10^5^ cells were seeded in 24-well plates and stimulated with Dynabeads™ Mouse T-Activator CD3/CD28 (Thermo Fisher Scientific). Cells were cultured in complete RPMI 1640 media, supplemented with either 1.5 μg/mL of mIL12 or an equivalent volume of PBS. After 48 h of incubation, levels of IFNγ were quantified using a Mouse IFNγ Precoated ELISA Kit (Dakewe, Shenzhen, China), according to the manufacturer’s instructions.

### 2.6. Flow Cytometric Analysis

Tumors were collected and subjected to mechanical and enzymatic dissociation. Tumors were mechanically dissected into small fragments and incubated with 0.5 mg/mL DNase I (Merck Sigma-Aldrich) and 0.5 mg/mL Collagenase Type IV (Thermo Fisher Scientific) in RPMI-1640 medium (Thermo Fisher Scientific) supplemented with 2% FBS. The dissociation was conducted in a shaker at 37 °C, 180 rpm, for 60 min. The sample was then passed twice through a 70 µm nylon cell strainer and washed with RPMI-1640 medium. After RBC lysis for 5 min at room temperature, the cells were washed with PBS and resuspended in FACS buffer for flow cytometric analysis. For surface staining, cells were plated at 2 × 10^6^ cells per 50 µL PBS in 1.5 mL tubes and stained with specific antibodies in FACS buffer for 30 min on ice. Following two washes in FACS buffer, cells were resuspended in 300 µL FACS buffer for flow cytometric analysis with a BD LSRFortessa X-20.

### 2.7. Antibody

The agents used for flow cytometry are listed below: anti-mouse CD45 BV395 (#564279, BD), anti-mouse CD3 PerCP-Cy5.5 (#100218, Biolegend), anti-mouse CD3 APC-Cy7 (#100222, Biolegend), anti-mouse CD3 AF700 (#100215, Biolegend), anti-Mouse CD4 FITC (#100406, Biolegend), anti-mouse CD8a BV605 (#100744, Biolegend), anti-Mouse PD-1 BV785 (#135225, Biolegend), anti-mouse CD152 APC (#106309, Biolegend), anti-Mouse CD62L APC-Cy7 (#104428, Biolegend), anti-mouse CD44 PE-Cy7 (#103030, Biolegend), anti-Mouse CD11b BV650 (#101239, Biolegend), anti-mouse CD11c BV421 (#117343, Biolegend), anti-Mouse I-A/I-E BV785 (#107645, Biolegend), anti-mouse XCR1 PerCP-Cy5.5 (#148208, Biolegend), anti-mouse CD103 PE (121406, Biolegend), Zombie Aqua™ Fixable Viability Kit (#423102, Biolegend). The anti-mouse CTLA-4 (#BE0032, Bio X Cell) and anti-mouse PD-1 (BE0273, Bio X Cell) all were purchased from Bioxcell, Inc (Bioxcell, West Lebanon, NH, USA).

### 2.8. Tumor Inoculation and Treatments

Murine syngeneic tumor models using MC38, CT26, and EMT6 cells involved subcutaneous inoculation of 5 × 10^6^ cells in 100 µL PBS into the right flank of C57BL/6 or Balb/c mice. For the B16-F10 model, 1 × 10^6^ cells in 100 µL PBS were used. Treatments commenced when tumors reached a predetermined volume, with either PBS, mIL12, or immune checkpoint inhibitors administered intraperitoneally. Tumor dimensions were measured every three days using electronic calipers, and the volume was calculated using the formula: Volume = Length × Width^2/2.

### 2.9. Animal

Balb/c and C57BL/6J mice were sourced from Shanghai Slack Laboratory Animal Co. and maintained under specific pathogen-free conditions with controlled temperature and humidity. Animal use was approved by the Institutional Animal Care and Use Committee at Xiamen University (XMULAC20150016) 2015,02.

### 2.10. Statistical Analysis

Flow cytometry data were analyzed using FlowJo v10.10.0. All other statistical analyses were conducted using GraphPad Prism v10.1. Details of the statistical tests and sample sizes are provided in each figure legend. Differences were considered statistically significant at *p* < 0.05. Significance in figures is indicated as * *p* < 0.05, ** *p* < 0.01, *** *p* < 0.001, **** *p* < 0.0001; ns indicates non-significant.

## 3. Results

### 3.1. Generation and Activity Assessment of mIL12 Cytokine

To evaluate the activity of mIL12, we initially engineered a recombinant gene encoding mIL12, consisting of two subunits, P40 and P35, linked by a flexible linker (G_4_S)_3_ and tagged with a His-tag at the C-terminus (Figure 1A). The recombinant mIL12 cytokine was expressed in 293F cells and analyzed by SDS-PAGE, which confirmed that the molecular weight of mIL12 was approximately 75 kDa, aligning with the theoretical value (Figure 1B). To assess its functionality, mIL12 significantly enhanced the release of mouse interferon-gamma (mIFN-γ) from mouse splenocytes in the presence of anti-mouse CD3 and anti-mouse CD28 antibodies (Figure 1C). Furthermore, to investigate the in vivo effects of mIL12 on tumor progression, MC38-bearing C57BL/6J mice were treated with PBS, an IL-12 inhibitor (anti-mouse IL-12), and mIL12. Results indicated that blocking IL-12 accelerated tumor progression, whereas mIL12 treatment markedly inhibited tumor growth (Figure 1D,E). These findings demonstrate the successful expression and functional verification of recombinant mIL12 in contributing to tumor suppression.

### 3.2. Potent Control of Tumor Growth by mIL12

The antitumor efficacy of mIL12 was assessed in xenograft tumor models using CT26, MC38, B16-F10, and EMT6 cells, subcutaneously established in mice (Figure 2A,D,G,J). Treatments with PBS and varying doses of mIL12 (0.01 mpk, 0.1 mpk, 1 mpk) commenced when tumor volumes reached approximately 151 mm^3^. Treatment with 0.1 mpk and 1 mpk doses resulted in complete tumor eradication in mice bearing CT26 tumors (Figure 2B). In models with immune checkpoint resistance, mIL12 significantly curtailed growth in MC38 tumors at all tested doses (Figure 2E). Furthermore, in the immunologically cold melanoma model (B16-F10), 0.1 mpk and 1 mpk of mIL12 demonstrated a strong antitumor effect (Figure 2H). Similar significant antitumor responses were observed in EMT6 tumor-bearing models following mIL12 treatment (Figure 2H,K). Crucially, no weight loss was observed in mice during the treatment with mIL12 (Figure 2C,F,I,L), underscoring its therapeutic potency across various tumor types.

### 3.3. Reshaping of the Tumor Immune Microenvironment by mIL12 and Upregulation of Immune Checkpoints

To elucidate the antitumor mechanism of mIL12, flow cytometry analysis was conducted on tumor samples from MC38-bearing mice post-treatment (Figure 3A). There was a notable increase in tumor-infiltrating CD3^+^ T cells, CD4^+^ T cells and CD8^+^ T cells (Figure 3B–D and Appendix A), indicating that mIL12 treatment significantly promotes T lymphocyte infiltration. Additionally, a marked presence of cDC1 and CD103^+^cDC1 cells was observed in the tumor microenvironment following mIL12 therapy, suggesting these dendritic cells enhance the T cell-mediated antitumor immune response (Figure 3E,F and Appendix A). Notably, the immunosuppressive molecules CTLA−4 and PD-1 were found to be upregulated in tumor-infiltrating T cells (Figure 3G,H and Appendix A), suggesting that while mIL12 reshapes the tumor immune environment and bolsters T cell responses, the upregulation of these molecules may limit therapeutic efficacy. Thus, combining mIL12 with PD-1 or CTLA-4 inhibitors could potentially enhance its antitumor effects.

### 3.4. Enhanced Antitumor Efficacy of mIL12 Combined with Immune Checkpoint Inhibitors

Immune checkpoint inhibitors are currently among the most effective strategies in tumor immunotherapy. To further explore the therapeutic potential of mIL12 in combination with these inhibitors, we employed the MC38 tumor-bearing model to assess the antitumor efficacy of mIL12 combined with PD-1 inhibitors versus monotherapy (Figure 4A). Results indicated that the antitumor effects of the combination treatment were superior to those observed with either mIL12 or PD-1 monotherapy alone (Figure 4B,D). Additionally, subsequent experiments with CTLA-4 inhibitors showed an even more pronounced antitumor effect in the combined treatment group (Figure 4E,F,H). Importantly, no significant body weight loss was observed in mice during the treatments (Figure 4C,G), underscoring the potential of combining mIL12 with immune checkpoint inhibitors to elicit a robust antitumor response without adverse effects. To investigate whether the combination of mIL12 and ICIs induces central memory T cell production, we conducted a comparison with monotherapy in an MC38 tumor model (Figure 4I). Significantly, our results demonstrated a notable increase in central memory T cells when mIL12 was combined with amPD-1 (Figure 4J and Appendix A). These findings suggest that the combination of mIL12 and PD-1 inhibitors achieves a more favorable therapeutic effect than either drug alone and is particularly beneficial for promoting the generation of memory T cells.

## 4. Discussion

In this study, we successfully expressed the mIL12 cytokine through eukaryotic recombination and confirmed its purity and molecular weight via SDS-PAGE. Our results demonstrated that mIL12 induced elevated levels of mouse IFN-γ in vitro. Experiments revealed that blocking IL-12 accelerated tumor growth in vivo, whereas treatment with mIL12 significantly controlled tumor immune responses. This suggests that IL-12 plays a critical role in the antitumor immune response, as evidenced by the activity of the mIL12 we produced. To further explore the antitumor efficacy of mIL12, we administered various doses to assess its therapeutic potential against several refractory tumor models. The antitumor effects of mIL12 appeared dose-dependent. Mechanistically, our study showed that mIL12 significantly enhanced the infiltration of CD4^+^ T cells, CD8^+^ T cells, cDC1, and CD103^+^cDC1 cells into the tumor microenvironment. Concurrently, an upregulation of the immune checkpoints PD-1 and CTLA-4 was observed. Notably, combining mIL12 with inhibitors of PD-1 and CTLA-4 yielded a superior antitumor response compared to monotherapy. Particularly, mIL12 combined with a PD-1 inhibitor markedly increased the population of tumor-infiltrating central memory T cells (T_CM_).

Our in vivo results consistently demonstrated a significant antitumor effect with mIL12 treatment, while IL-12 blockade led to enhanced tumor progression compared to control (PBS). Previous research indicates that the efficacy of PD-1 inhibitors is diminished following IL-12 blockade, highlighting the indispensable role of IL-12 in mediating tumor immunity [12]. The interaction between T cells and dendritic cells (DCs) within the cancer-immune cycle is crucial for a robust immune response. Enhanced infiltration of T and DC cells helps counteract tumor-induced immunosuppression, with cDC1 cells being essential for activating CD8^+^ T cell-driven antitumor responses [33,34,35]. Our flow cytometry analysis revealed a significant increase in tumor-infiltrating T and DC cells, indicating that mIL12 monotherapy can shift the tumor’s microenvironment from an immunosuppressive to a pro-inflammatory state. Furthermore, the expression of PD-1 and CTLA-4 on T cells suggests that combining immune checkpoint inhibitors (ICIs) could amplify the antitumor effects. This finding aligns with studies showing enhanced antitumor activity when IL-12 is combined with dual PD-1 and CTLA-4 blockade [36]. However, clinical studies have reported severe adverse effects with such combinations, leading us to explore mIL12 in combination with PD-1 or CTLA-4 inhibitors separately [37,38]. Both strategies showed more effective tumor control compared to monotherapy. Intriguingly, the combination of PD-1 blockade with mIL12 significantly increased central memory CD8^+^ T cells, known for their potent and durable antitumor responses [39]. The presence of memory T cells has been correlated with a more favorable prognosis in cancer patients. Long-lived memory CD8^+^ T cells play a crucial role in tumor immunity as they possess the ability to exert a durable antitumor response [11]. In this study, our results first provided evidence that a combination of IL-12 and PD-1 blockades can augment tumor-infiltrating memory CD8^+^ T cells.

While this study highlights the potential of mIL12 to enhance immune checkpoint blockade therapy, there are limitations. The potent antitumor activity of IL-12 could induce side effects, and further toxicological data are needed. Additionally, the substantial increase in memory CD8^+^ T cells with mIL12 and PD-1 inhibitor combination warrants further investigation into the mechanisms of memory T cell formation.

Given the potential toxicity associated with IL-12 in clinical settings, it is crucial to explore flexible utilization strategies. IL-12 can be employed as a standalone therapeutic or integrated into combination therapies with PD-1 inhibitors, allowing for precise regulation and dose adjustments. To minimize systemic toxicity, various administration methods, such as in situ delivery and subcutaneous injections, can be considered. Moreover, innovative approaches like the engineering of immune cytokine fusion proteins could be pursued. These proteins would combine IL-12 with antibodies that bind high-affinity PD-1^+^ cells, potentially achieving a synergistic antitumor effect by simultaneously blocking the PD-1 pathway and activating IL-12 signaling.

In conclusion, recombinant mIL12 significantly suppressed tumor growth in multiple refractory tumor models. The combination of mIL12 with PD-1 or CTLA-4 blockade showed superior antitumor efficacy compared to monotherapy, with the combination of mIL12 and PD-1 blockade notably enhancing the number of memory CD8^+^ T cells. These findings offer promising directions for optimizing immune checkpoint blockade therapy and developing strategies for curative cancer immunotherapy.

## Figures and Tables

**Figure 1 antibodies-13-00094-f001:**
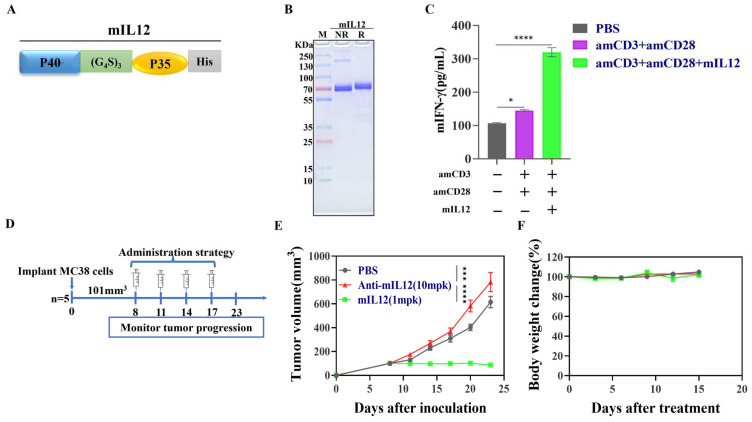
Expression and characterization of mIL12. (**A**) Genetic map illustrating the coding gene for mIL12. (**B**) SDS-PAGE analysis of mIL12 under reducing (R) and non-reducing (NR) conditions, with molecular weight (MW) markers shown. (**C**) mIFN-γ levels in the culture medium of splenocytes from C57BL/6 mice, stimulated ex vivo with PBS, anti-mouse CD3+ anti-mouse CD28 (amCD3+amCD28), or amCD3+amCD28+mIL12. mIFN-γ levels were quantified using an ELISA assay, and the data are representative of three independent experiments. (**D**) Treatment regimen for MC38 tumor−bearing C57BL/6J mice, which received PBS, mIL12 (1 mg/kg), or anti-mIL12 (10 mg/kg) therapy (n = 5 per group). (**E**) Growth curves of tumors and (**F**) body weight changes in C57BL/6J mice. Data are presented as means ± SEM. Statistical analysis was conducted using one−way ANOVA for panel (**C**) and two-way ANOVA for panel (**E**). Significance levels are indicated as * *p* < 0.05; *** *p* < 0.001; **** *p* < 0.0001.

**Figure 2 antibodies-13-00094-f002:**
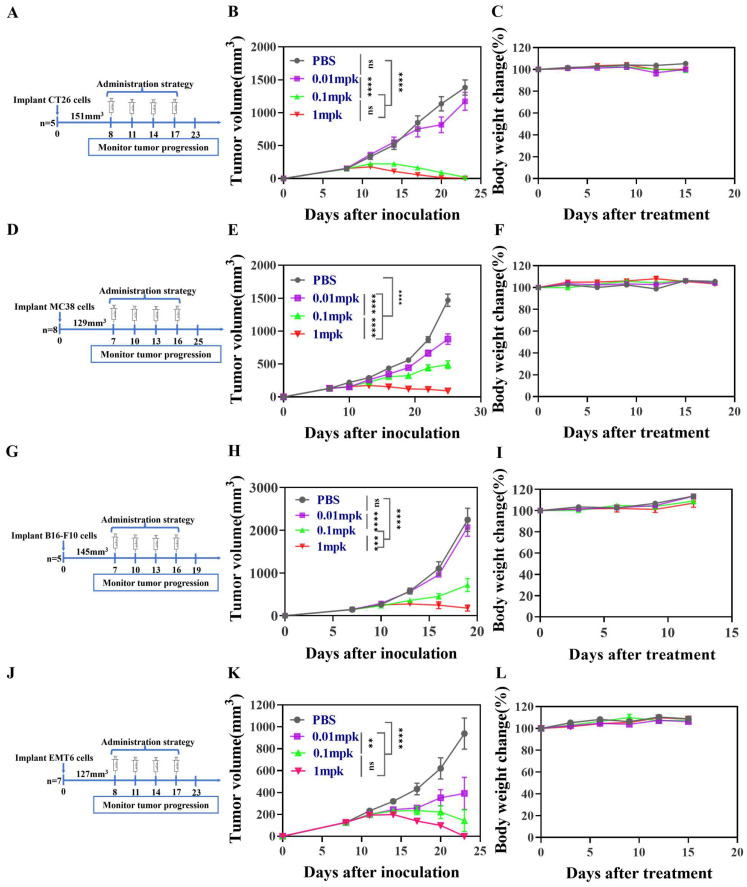
Dose-dependent antitumor efficacy of mIL12 in mouse tumor models. Treatment regimens for tumor-bearing mice: CT26 (n = 8), MC38 (n = 8), B16-F10 (n = 5), and EMT6 (n = 7) (Panels (**A**,**D**,**G**,**J**). Mice were treated intraperitoneally with PBS or mIL12 at doses of 0.01 mg/kg, 0.1 mg/kg, or 1 mg/kg every three days. Tumor growth curves (Panels (**B**,**E**,**H**,**K**) and body weight changes (Panels (**C**,**F**,**I**,**L**) of the mice are shown. Data are presented as means ± SEM. Statistical analyses for tumor growth (Panels (**B**,**E**,**H**,**K**) were conducted using two-way ANOVA. Significance levels are denoted as ns (no significant difference), ** *p* < 0.01, *** *p* < 0.001, and **** *p* < 0.0001.

**Figure 3 antibodies-13-00094-f003:**
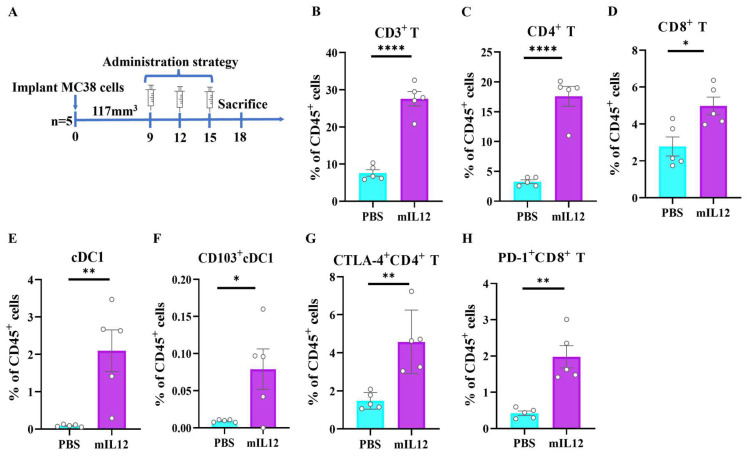
Reshaping of the tumor immune microenvironment and upregulation of immune checkpoints by mIL12. (**A**) Schematic of the treatment strategy for the MC38 tumor model. Tumors from mice treated with mIL12 were dissociated and analyzed by flow cytometry. (**B**–**F**) Proportions of tumor-infiltrating T cells and dendritic cells (DCs) among live CD45^+^ cells. (**G**) Percentage of CTLA-4^+^CD4^+^ T cells and (**H**) PD-1^+^CD8^+^ T cells within the live CD45^+^ cell population. Data are presented as means ± SEM. Statistical analyses for panels (**B**–**H**) were performed using unpaired two-tailed Student’s *t*-test. Significance levels are indicated as * *p* < 0.05, ** *p* < 0.01, and **** *p* < 0.0001.

**Figure 4 antibodies-13-00094-f004:**
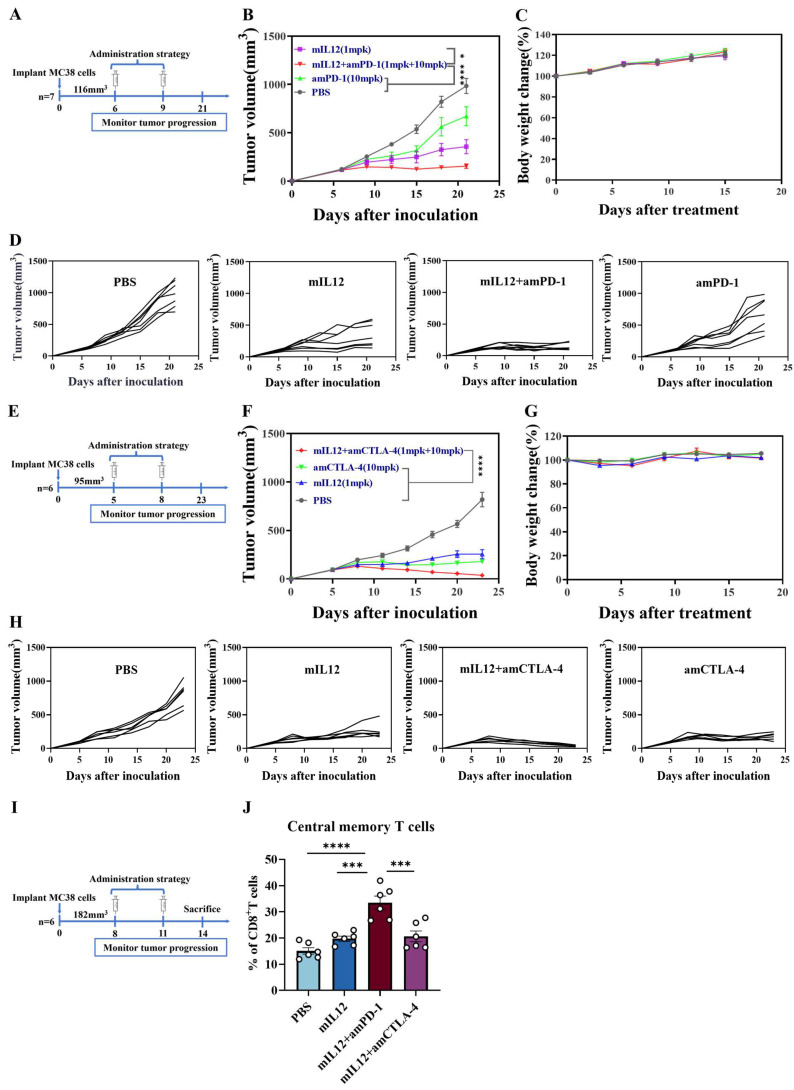
Enhanced antitumor efficacy of mIL12 combined with PD-1 and CTLA-4 immune checkpoint inhibitors. (**A**) Treatment scheme comparing the therapeutic efficacy of mIL12 combined with anti-mouse PD-1 (amPD-1) antibody versus monotherapy. Mice bearing MC38 tumors received treatments with PBS, mIL12, amPD-1, or a combination of amPD-1 with mIL12. (**B**) Tumor growth curves and (**C**) body weight changes for monotherapy and combination therapy groups. (**D**) Individual tumor growth curves for each treatment group (n = 7 per group). (**E**) Treatment scheme comparing the therapeutic efficacy of mIL12 combined with anti-mouse CTLA-4 (amCTLA-4) antibody versus monotherapy. Mice bearing MC38 tumors received PBS, mIL12, amCTLA-4, or a combination of amCTLA-4 and mIL12. (**F**) Tumor growth curves and (**G**) body weight changes for monotherapy and combination therapy groups. (**H**) Individual tumor growth curves for each treatment group (n = 6 per group). (**I**) Schematic of the treatment strategy for the MC38 tumor model. Tumors from mice treated with PBS, mIL12, mIL12+amPD-1 and mIL12+amCTLA-4 were dissociated and analyzed by flow cytometry. (**J**) The proportion of tumor-infiltrating central memory T cells. Data are presented as means ± SEM. Statistical analyses for (**B**,**F**) were performed using two-way ANOVA. Statistical analyses for (**J**) were performed using one-way ANOVA. Significance levels are indicated as * *p* < 0.05, *** *p* < 0.001, **** *p* < 0.0001.

## Data Availability

The datasets generated and/or analyzed in this study are available from the corresponding author upon reasonable request.

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
