# Peer review of "Enhancing Tumor Immunity with IL-12 and PD-1 Blockade: A Strategy for Inducing Robust Central Memory T Cell Responses in Resistant Cancer Model"

_2073-4468, 2024, doi:10.3390/antib13040094_

Round 1
Reviewer 1 Report
Comments and Suggestions for Authors
The immuntherapy represents today more or less the cornerstone ot the treatment in oncology. The vast majority of the puiblished research were for inhibitors receptors which are blocked for increasing the activitity of LyT, but less for stimulating the imune activity. This article is in this particular area, which is meaningful.
Maybe a short description of the interpherence with secretion or inhibitors of IL12 could be helpful.
Author Response
Comments and Suggestions for Authors
The immuntherapy represents today more or less the cornerstone on the treatment in oncology. The vast majority of the puiblished research were for inhibitors receptors which are blocked for increasing the activitity of LyT, but less for stimulating the imune activity. This article is in this particular area, which is meaningful.
Response: We sincerely appreciate the reviewer's positive feedback and valuable comments on our manuscript. Your constructive suggestions have been instrumental in enhancing the quality of our work. In response to your recommendations, we have undertaken significant revisions to the manuscript. We believe these changes have strengthened our presentation and addressed the concerns raised during the review process.

Reviewer 2 Report
Comments and Suggestions for Authors
Dear Authors:
The manuscript by Chen et al. entitled “Enhancing tumor immunity with IL-12 and PD-1 blockade: A strategy for inducing robust central memory T cell responses in resistant cancer model” is well written in English. The study seems to be well planned out and the results are a significant contribution to the present knowledge in cancer immunotherapy. However, the manuscript can be improved by addition of several pieces of information.
Major Comments:
1. In the second paragraph of the introduction, line 44, add a sentence to begin the paragraph that would include the subject matter of the paragraph. It should tie IL-12 to ICI treatment.
2. In the methods section include information about purification of IL-12. Also, include an explanation of how purified IL-12 was employed to stimulate IFN-γ secretion by mouse splenocytes. Add a section on quantification of IL-12 and IFN-γ by ELISA to the methods section.
3. Add a section to the methods about how the cells were counted and how viability measurements were performed.
4. In the flow cytometry section indicate the company and model of the flow cytometer.
5. In the method section about SDS-PAGE, indicate how the proteins were prepared for running on the SurePAGE gels. Mention any homogenization process and protease inhibitors employed, etc.
6. In figure 1, the Y-axis label on graph E has the word “Volume” misspelled. Check all of the graphs in the manuscript to make sure that all labels are correct.
7. In figure 1, the Y-axis for graph F should indicate % body weight change.
8. In the results section about T-cell and dendritic cell infiltration into the tumor it would be helpful to present a few representative plots of the flow cytometry data. Indicate gating strategies and give percentages of cells of interest. This would indicate to the reader that the flow cytometry was performed correctly, and the data obtained is reliable. Although it does appear that the cytometry was well performed, it is always good to see the raw data.
9. In figure 4, there is no mention of graph J in the figure legend. Further, give an indication of which markers were used to identify central memory T cells, both CD4 and CD8 positive T cells.
1 In the discussion it is good to get a summary of the results of the study, but your results should be placed in the context of what others have discovered. Also, emphasize how your results add to the knowledge about cancer treatment. Indicate how further studies could be conducted that might lead to protocols toward incorporation of IL-12 into clinical treatment strategies.
1 Increase the size of the labeling slightly on all your graphs. It is very difficult to read the labels.
Author Response
The manuscript by Chen et al. entitled “Enhancing tumor immunity with IL-12 and PD-1 blockade: A strategy for inducing robust central memory T cell responses in resistant cancer model” is well written in English. The study seems to be well planned out and the results are a significant contribution to the present knowledge in cancer immunotherapy. However, the manuscript can be improved by addition of several pieces of information.
Response: We are grateful to the reviewer for the thorough examination of our manuscript and for the insightful comments provided. Your feedback has been invaluable in improving the quality of our work. In accordance with your suggestions, we have made substantial revisions to the manuscript, detailed as follows.
Major Comments:
- In the second paragraph of the introduction, line 44, add a sentence to begin the paragraph that would include the subject matter of the paragraph. It should tie IL-12 to ICI treatment.
Response: Thank you for your suggestion. To better connect IL-12 with immune checkpoint inhibitor (ICI) treatment in the context of our discussion, we have added a sentence at the beginning of the second paragraph of the introduction, now found in line 66-69.
- In the methods section include information about purification of IL-12. Also, include an explanation of how purified IL-12 was employed to stimulate IFN-γ secretion by mouse splenocytes. Add a section on quantification of IL-12 and IFN-γ by ELISA to the methods section.
Response: Thank you for your insightful recommendations. We have updated the methods section to include detailed information on the purification process of IL-12, as well as its use in stimulating IFN-γ secretion by mouse splenocytes. Additionally, we have added a subsection detailing the quantification of both IL-12 and IFN-γ using the ELISA method in line 139-145.
- Add a section to the methods about how the cells were counted and how viability measurements were performed.
Response: Thank you for your suggestion. In response, we have added detailed descriptions of the cell counting and viability measurement techniques to the methods section of our manuscript, ensuring clarity on our experimental procedures in line 120-125.
- In the flow cytometry section indicate the company and model of the flow cytometer.
Response: Thank you for your reminder. We have updated the methods section to include the specific company and model of the flow cytometer used in our experiments in line 157-158.
- In the method section about SDS-PAGE, indicate how the proteins were prepared for running on the SurePAGE gels. Mention any homogenization process and protease inhibitors employed, etc.
Response: Thank you for your suggestion. We have revised the methods section to include detailed information on how the proteins were prepared for SDS-PAGE, including any homogenization processes and the use of protease inhibitors before running on SurePAGE gels in line 132-137.
- In figure 1, the Y-axis label on graph E has the word “Volume” misspelled. Check all of the graphs in the manuscript to make sure that all labels are correct.
Response: We apologize for this oversight and appreciate your attention to detail. We have corrected the spelling of "Volume" in Figure 1 and have thoroughly reviewed and updated all other graphs in the manuscript to ensure that all labels are accurate. Thank you for bringing this to our attention.
- In figure 1, the Y-axis for graph F should indicate % body weight change.
Response: We apologize for the oversight in the labeling of Figure 1, graph F. Thank you for pointing it out. We have corrected the Y-axis to accurately reflect "% body weight change."
- In the results section about T-cell and dendritic cell infiltration into the tumor it would be helpful to present a few representative plots of the flow cytometry data. Indicate gating strategies and give percentages of cells of interest. This would indicate to the reader that the flow cytometry was performed correctly, and the data obtained is reliable. Although it does appear that the cytometry was well performed, it is always good to see the raw data.
Response: Thank you for your constructive suggestion. We have included representative flow cytometry plots in the supplementary figures to provide clear visual evidence of our gating strategies and the percentages of the cells of interest. We appreciate your recommendation to enhance the transparency and credibility of our results.
- In figure 4, there is no mention of graph J in the figure legend. Further, give an indication of which markers were used to identify central memory T cells, both CD4 and CD8 positive T cells.
Response: We apologize for the oversight regarding graph J in Figure 4 and appreciate your attention in pointing this out. We have updated the figure legend to include a description of graph J in line 297. Additionally, we have detailed the flow cytometry gating strategy used to identify central memory T cells (Tcm), including both CD4 and CD8 positive T cells, in the supplementary figures.
- In the discussion it is good to get a summary of the results of the study, but your results should be placed in the context of what others have discovered. Also, emphasize how your results add to the knowledge about cancer treatment. Indicate how further studies could be conducted that might lead to protocols toward incorporation of IL-12 into clinical treatment strategies.
Response: Thank you for your valuable suggestion. We have revised the discussion section to place our results within the broader context of existing research. Notably, our study builds on prior findings by demonstrating that the combination of IL-12 with PD-1 inhibitors enhances the proportion of central memory T cells, an aspect not previously documented. We have also differentiated our findings from those focusing on myeloid cells and the use of PD-1/CTLA-4 dual immune checkpoint therapy to mitigate immunosuppression. Furthermore, we have expanded our discussion to include potential future studies, outlining how our results could inform clinical protocols that integrate IL-12 into cancer treatment strategies more effectively in line 344-351.
- Increase the size of the labeling slightly on all your graphs. It is very difficult to read the labels.
Response: Thank you for your feedback regarding the legibility of our graph labels. We have increased the size of the labeling on all graphs to ensure they are easier to read. We appreciate your suggestion and have made sure that these changes enhance the clarity of our figures.

Reviewer 3 Report
Comments and Suggestions for Authors
The authors present tumor mouse models treated concomitantly with IL-12 and checkpoint inhibitors to show that the combination of the two is efficient in removing tumor burden. The experiments and data are clear. However, some data are missing.
The Introduction should be expanded. Further information about IL-12 and its impact upon the immune system should be mentioned, including the Th1 skewing, leading not only to IFNγ, but also TNFα production. Same goes for the checkpoint inhibitors.
T central memory cells should also be described in terms of phenotype and differentiation, as they are mentioned in the abstract.
Many more details should be offered regarding previous studies involving IL-12 administration and the important/multiple side effects that effectively lead to the termination of IL-12 based clinical trials by the FDA. These studies were then resumed only by a small number of centers.
Line 73: please detail what are the Expi 293F cells are.
Should this technique was already used, you may want to mention the publication
Line 85: I don’t think the term „plated” is appropriate for the incubation in a tube
Line 102: please offer details regarding the MC38, CT26, and EMT6 cells
Line 104: Just as a curiosity, why the melanoma cells B16F10 required 5 times less cells? Was this optimized for the current manuscript, or it has been published before?
Line 181: no flow cytometry data are offered regarding the CTLA-4 and PD-1 up-regulation
Line 208: no flow cytometry data are offered regarding the central memory T cells, neither in terms of phenotype, nor in terms of actual data showing their increase in numbers, as compared to controls. Further, even though it is later mentioned that rather the CD8+ cells were of interest for the authors, there are no details if both CD4+ and CD8+ cells were investigated.
Discussion: I think it would be important to stress that you opted for IL-12 inoculation, which might actually explain the lack of general side effects, assessed by measuring the body weight.
Further, it would be worthwhile commenting on how your approach might be translated to human subjects.
Author Response
The authors present tumor mouse models treated concomitantly with IL-12 and checkpoint inhibitors to show that the combination of the two is efficient in removing tumor burden. The experiments and data are clear. However, some data are missing.
Response: We sincerely appreciate the reviewer's thorough review and valuable comments on our manuscript. Your insights have significantly contributed to the improvement of our document. Following your suggestions, we have made extensive revisions to the manuscript, which are outlined below. These changes aim to enhance the clarity, accuracy, and overall quality of our work.
1、The Introduction should be expanded. Further information about IL-12 and its impact upon the immune system should be mentioned, including the Th1 skewing, leading not only to IFNγ, but also TNFα production. Same goes for the checkpoint inhibitors.
Response: We sincerely thank the reviewer for the insightful suggestions. In response, we have expanded the Introduction to provide additional information on IL-12 and its effects on the immune system, including its role in Th1 skewing and the subsequent production of IFNγ and TNFα. We have also added further details about checkpoint inhibitors, as suggested, in lines 74-86 of the revised manuscript.
2、T central memory cells should also be described in terms of phenotype and differentiation, as they are mentioned in the abstract.
Response: Thank you for your valuable suggestion. We have added a detailed description of T central memory cells, including their phenotype and differentiation, in the manuscript, as suggested in lines 56-65.
3、Many more details should be offered regarding previous studies involving IL-12 administration and the important/multiple side effects that effectively lead to the termination of IL-12 based clinical trials by the FDA. These studies were then resumed only by a small number of centers.
Response: Thank you for your insightful suggestion. In accordance with your recommendation, we have expanded the manuscript to include additional details on previous studies involving IL-12 administration, highlighting the significant side effects that led to the termination of IL-12-based clinical trials by the FDA. We have also mentioned the circumstances under which these studies were resumed by a limited number of centers. This information has been added in lines 93-100 of the revised manuscript.
4、Line 73: please detail what are the Expi 293F cells are. Should this technique was already used, you may want to mention the publication
Response: Thank you for your suggestion. We have added a detailed description of Expi293F cells in the methods section, including their characteristics and relevance to our study. Additionally, we have cited relevant publications where this technique has been previously used, as mentioned in lines 128-129 of the revised manuscript.
5、Line 85: I don’t think the term “plated” is appropriate for the incubation in a tube
Response: Thank you for your suggestion. We have revised the wording in the manuscript to replace the term "plated" with a more appropriate term for incubation in a tube, as reflected in line 149-151.
6、Line 102: please offer details regarding the MC38, CT26, and EMT6 cells
Response: Thanks for the reviewer’s advice. We have added the description in the manuscript as suggested in line 112-116.
7、Line 104: Just as a curiosity, why the melanoma cells B16F10 required 5 times less cells? Was this optimized for the current manuscript, or it has been published before?
Response: Thank you for your interest in this aspect of our study. The B16-F10 cell line is known for its highly malignant nature and rapid growth, making it a challenging tumor model. Inoculating 5 million cells would have resulted in tumors that exceeded ethical volume limits before the completion of the administration cycle. Therefore, we optimized our protocol by reducing the number of inoculated cells to ensure adherence to ethical guidelines. This adjustment was specifically tailored for our study, and no prior publication detailing these modifications exists.
8、Line 181: no flow cytometry data are offered regarding the CTLA-4 and PD-1 up-regulation
Response: We apologize for the oversight and thank the reviewer for bringing this to our attention. We have now included the flow cytometry data illustrating the up-regulation of PD-1 and CTLA-4, which can be found in Figures 3G and 3H in line 253.
9、Line 208: no flow cytometry data are offered regarding the central memory T cells, neither in terms of phenotype, nor in terms of actual data showing their increase in numbers, as compared to controls. Further, even though it is later mentioned that rather the CD8+ cells were of interest for the authors, there are no details if both CD4+ and CD8+ cells were investigated.
Response: We apologize for the oversight and appreciate the reviewer’s careful review of our manuscript. Data on central memory T cells (Tcm) are now presented in Figure 4J in line 297, and the flow cytometry gating strategy used to identify these cells is detailed in Supplementary Figure S2. Our study primarily focused on the analysis of anti-tumor effector cells, specifically CD8+ T cells, and did not include extensive comparisons with CD4+ T cells. We have clarified this focus in the revised manuscript.
10、Discussion: I think it would be important to stress that you opted for IL-12 inoculation, which might actually explain the lack of general side effects, assessed by measuring the body weight.
Response: Thank you for your insightful comment. We agree that our choice of IL-12 inoculation may have contributed to the absence of general side effects, as indicated by stable body weight measurements. However, we recognize that monitoring body weight is only one aspect of assessing drug safety. Cytokine therapies, including IL-12, require more comprehensive and rigorous in vivo safety evaluations. In future studies, we plan to expand our safety assessments to include detailed serological and histological analyses to provide a more thorough evaluation of potential side effects.
11、Further, it would be worthwhile commenting on how your approach might be translated to human subjects.
Response: Thank you for your valuable suggestion. We have added a discussion on the potential translation of our approach to human subjects in the manuscript, as suggested, in lines 344-351. This addition highlights how our findings might inform future clinical strategies and applications.

Round 2
Reviewer 3 Report
Comments and Suggestions for Authors
Thank you for considering and addressing all the obseravtions
Author Response
We are grateful to the editorial team and all the reviewers for their comments and suggestions, which have made our submitted article more comprehensive and complete.